# Mycorrhized Wheat Plants and Nitrogen Assimilation in Coexistence and Antagonism with Spontaneous Colonization of Pathogenic and Saprophytic Fungi in a Soil of Low Fertility

**DOI:** 10.3390/plants11070924

**Published:** 2022-03-29

**Authors:** Catello Di Martino, Valentina Torino, Pasqualino Minotti, Laura Pietrantonio, Carmine Del Grosso, Davide Palmieri, Giuseppe Palumbo, Thomas W. Crawford, Simona Carfagna

**Affiliations:** 1Department of Agriculture, Environmental and Food Sciences, University of Molise, Via De Sanctis, 86100 Campobasso, Italy; v.torino@studenti.unimol.it (V.T.); minottipasqualino@gmail.com (P.M.); c.delgrosso2@studenti.unimol.it (C.D.G.); davide.palmieri@unimol.it (D.P.); palumbo@unimol.it (G.P.); 2MS Biotech SpA, c.da Piane di Larino, 35, 86035 Larino, Italy; laura.pietrantonio@msbiotech.net; 3Global Agronomy, LLC, Marana, AZ 85658, USA; globalagronomy@gmail.com; 4Dipartimento di Biologia, Università degli Studi di Napoli Federico II, 80126 Napoli, Italy; simona.carfagna@unina.it

**Keywords:** amino acids, nitrate reductase, glutamine synthetase, plants mycorrhized, dark septate

## Abstract

The aim of the work was to study the biological interference of the spontaneous colonization of pathogenic and saprophytic endophytes on the nitrogen assimilation of mycorrhized wheat plants cultivated in soils deficient in N and P. The nitrogen assimilation efficiency of mycorrhized plants was determined by measuring the activities of nitrate reductase assimilatory and glutamine synthetase enzymes and free amino acid patterns. Mycorrhizal plants at two different sites showed an assimilative activity of nitrate and ammonium approximately 30% greater than control plants. This activity was associated with significant increases in the amino acids Arg, Glu Gln and Orn in the roots where those amino acids are part of the inorganic nitrogen assimilation of mycorrhizal fungi. The nutrient supply of mycorrhizal fungi at the root guaranteed the increased growth of the plant that was about 40% greater in fresh weight and 25% greater in productive yield than the controls. To better understand the biological interaction between plant and fungus, microbiological screening was carried out to identify colonies of radicular endophytic fungi. Fourteen fungal strains belonging to nine different species were classified. Among pathogenic fungi, the genus *Fusarium* was present in all the examined roots with different frequencies, depending on the site and the fungal population present in the roots, providing useful clues regarding the principle of spatial conflict and fungal spread within the root system.

## 1. Introduction

About 80% of terrestrial plant roots are closely associated with mycorrhizal fungi [1], and many aspects of the physio-ecological roles played by these mycorrhizal fungi, such as plant nutrition, soil biology and soil chemistry, are well described [2]. The cytological interaction between fungus and root occurs at the interface between the plasma membranes of the arbuscular and cortical cells. Through these contact surfaces, nutritional exchanges take place between the plant that supply the fungus with carbonaceous skeletons and the fungus that transfers the mineral nutrients to the plant [3,4]. It has been reported that phosphate deficiency in soil stimulates mycorrhizal colonization [5]. The suppression of mycorrhizae arises from the high concentration of P and from other mineral nutrients present in the soil. In addition, a limitation of nitrogen in the soil also stimulates root colonization by mycorrhizal fungi [6]. Moreover, mycorrhizal fungi are implicated in absorbing inorganic nitrogen from the soil and moving it to the roots to be partly assimilated by the root and partly by the leaves into organic nitrogen compounds. In both the roots and the leaves, the reduction of nitrate and the subsequent assimilation of ammonium require a significant energy quotient to generate to organic N compounds through the consumption of the reduction equivalents of nicotinamide adenine dinucleotide (NAD(P)H) and adenosine triphosphate (ATP).

In mycorrhizae, nitrate is absorbed by the extra-radical hyphae and is then reduced to ammonium by nitrate reductase assimilatory (NRA) enzyme via the fugin metabolic pathway [7,8]. Ammonium absorbed from the soil or generated by nitrate reduction is the nitrogen substrate of two metabolic pathways: Glutamine synthetase (GS) and Glutamine-2-oxoglutarate aminotransferase (GOGAT) [9]. The glutamate (Glu) produced is quickly converted to arginine (Arg) that is then transferred into the intraradical mycelium. The disintegration of Arg releases NH_4_^+^ from the arbuscules to the root cortex cells, which assimilate it through the GS/GOGAT pathway [10,11].

The roots of plants and mycorrhizae can also be colonized by the endophytic fungi, such as *Tricoderma* species, and dark septate fungi present in the rhizosphere. Interactions between plants and endophytic fungi in the roots do not always benefit the host. It is essential to clarify some fundamental concepts of phytopathology. When a plant and another organism come into contact, the interaction can be beneficial, harmless or harmful. Beneficial interactions, such as mutualism (each organism benefits from the association) and commensalism (all organisms in association are mutually beneficial), can be a source of great opportunity. Organisms involved in mutualism or commensalism can promote plant growth, induce abiotic stress resistance or make the plant more resistant to a disease. Harmful interactions include, on the other hand, competition, antagonism and parasitism [12]. These negative interactions can stress and damage cultivated plants and can negatively affect the treatments applied using biological agents. Therefore, considering the wide variety of species of endophytic fungi present in the soil, mycorrhizae can interact with phytopathogenic, saprophytic or symbiotic fungal colonies in the roots of the host plant [13,14].

Dark septate endophytes (DSE) are a group of endophytic fungi broadly classified as conidial or sterile septate fungal endophytes that form melanized structures, such as inter- and intracellular hyphae and microsclerotia in the plant roots, and DSE are believed to represent primary non-mycorrhizal root-inhabiting fungi [15]. Wild plant species may live in symbiosis with a unique and rich mycoflora, including DSE that may have been lost during the breeding of the cultivars used in agriculture [16,17]. However, some agronomic crops, such as grasses and cereal roots, are often colonized by DSE. Frequently DSE occur together with mycorrhizal fungi, such as arbuscular, ericoid, orchid and ectomycorrhiza [18]. Colonization by fungi through their hyphae enables host plants to acquire soluble mineral nutrients not otherwise available [18]. The relatively small diameter of DSE hyphae, with respect to the diameter of the root, allows roots to indirectly penetrate soil micropores and acquire resources from a volume of soil otherwise impenetrable to the plant’s roots. It has been shown that *Piriformospora indica* colonizes the roots of many species of plants, favoring their growth [19,20].

In ways similar to mycorrhizal symbiosis, the symbiosis of *P. indica* with plant roots is characterized by a large intake of N [21,22]. In fact, Cruz et al. [23] observed that the absorption rate of NH_4_^+^ labelled with ^15^N by the extraradical mycelium was greater in the tomato–*P. indica* interaction than in tomato–*Glomus intraradices* interaction. There is some evidence that different kinds of root-associated fungi interact. For example, ectomycorrhizae and DSE strains together are associated with increased plant biomass that is more than with either alone [24]. Cereal mycorrhizae, an ancient symbiotic, are often found with DSE associated with the roots of wheat (*Triticum aestivum*), wild barley (*Hordeum brevisubulatum* and *Hordeum bogdanii*), soybean (*Glycine max*) and maize (*Zea mays*) [25]. Even if there is little evidence of mutualistic interaction between plants and DSE [26,27], there are indications that the ecophysiological contribution DSE can provide to the plant include stress tolerance and resistance to pathogens. In other words, the presence of DSE in plant roots could have a function very similar to a probiotic factor: the spores of DSE, once released, rapidly colonize the environment, occupying the ecological niche of other fungi and, by minimizing the carbon available in host rhizosphere environment, may prevent the development of other fungi. This indicates that when the soil is colonized by DSE, plants may be attacked less by fungal diseases of the root system. The relatively small diameter of DSE hyphae, especially when compared to the root diameter, allows the penetration of soil micropores and the acquisition of resources from a soil volume impenetrable to larger plant roots. As mentioned above, unlike mycorrhizal fungi, endophytes do not establish symbiosis with the host plant, but manage to live asymptomatically within the plant tissue. To protect the host plant from biotic and abiotic stresses, endophytic fungi produce different bioactive compounds, and, in turn, the endophytic fungi benefit from nutrition provided by the host plant [28]. This exchange of benefits makes endophytes a potential resource for the production of beneficial metabolites, such as antimicrobial, antiviral and insecticidal compounds, which are antioxidants and compounds antagonistic to some kinds of cancer [28]. In view of the results of emerging research, many bioactive compounds attributed in the past to the secondary metabolism of plants may be, instead, of fungal origin [28]. Among the wide range of endophytic fungi present in the soil, the phytopathogens of cultivars of agronomic interest, such as wheat, have been a continuing source of great damage to humans and their health. Unfortunately, the roots of wheat can become colonized by *Fusarium* species, often resulting in severe crop disease, the accumulation of secondary metabolites toxic to man and reduced yields. The fungal species mainly associated with *Fusarium* head blight (FHB) in Europe are *F. graminearum*, *F. avenaceum* and *F. poae* [29,30,31]. Less frequently isolated species are *F. tricinctum*, *F. sporotrichioides*, *F. equiseti*, *F. langsethiae* and *F. culmorum* [32,33,34]. The purpose of the present work is to identify the effects of the biological interaction between endophytes and mycorrhizal durum wheat roots on nitrogen metabolism and the yield of durum wheat (*Triticum durum* Desf. or *Triticum turgidum* subsp. d*urum* (Desf.) van Slageren) [35]) in fields with low rates of fertilization of phosphorus and nitrogen.

## 2. Results

### 2.1. Mycorrhization of Roots and Durum Wheat Development

The mycorrhizal colonization and durum wheat growth depicted in Figure 1 show the growth of mycorrhizal (M) and non-mycorrhizal control (NM) plants at two field sites. Spontaneous mycorrhization growth curves show a maximum value below 10% root length colonization (RLC) for NM particles under optimal soil fertilization for both sites A and B. Differently, M particles with low fertilization and mycorrhization growth curves show a maximum value of more than 50% (RLC) for both sites. These data show that the growth trend of the plants was influenced by the development of mycorrhizal colonies inside the root at both sites. Between 90 and approximately 130 days after sowing (DAS), i.e., during the phase of the beginning of stem elongation, the plants treated with mycorrhizae (M) grew approximately 30–40% less than the non-treated control plants (NM). At 130 DAS, i.e., during the beginning of the stem elongation stage, M and NM plants exhibited equivalent masses. Between 130 and 190 DAS, when root colonization was at the highest extent (reaching a maximum of 50 to 60% RLC), the shoots of M grew better than those of NM plants in both sites. In the range between 130 and 190 DAS, i.e., at the end of the raising stage and when the mycorrhization was maximal, the shoots of M plants had a greater mass than the shoots of NM plants, with a 25% and 35% increase for A and B sites, respectively.

### 2.2. Nitrogen Metabolism

In the present experiment, nitrogen in the soil was augmented with fertilizer containing 30% NO_3_^−^-N and 30% ureic N. Nitrogen fertilizer was scattered 200 kg ha^−1^ (NM) and 100 kg ha^−1^ (M) on the plants [36]. The activities of NRA and GS, crucial enzymes of N metabolism and assimilation, were studied in comparison to nitrogen sources and mycorrhiza–plant symbiosis (Figure 2). At 190 DAS in the control (NM) plants, the NRA activity was higher in leaves (Figure 2b) than in roots (Figure 2a). There was 27% and 30% more NRA activity at sites A and B, respectively. The values of NRA activity were greatest for plants treated with mycorrhizae (M) compared to NM plants in both roots and leaves at both sites. The largest increase in the NRA activity of M plants on NM plants was observed in the roots in both locations, with an average increase of 70% (Figure 2a). An analogous pattern was observed in leaves, but to a lesser extent in both locations with an average increase of 45% (Figure 2b).

It is particularly interesting to note that in NM plants, the specific activity of NRA was greater in the leaves than in the roots. On the contrary, in M plants, the percentage increase in NRA and its absolute values were higher in roots than in leaves. In NM plants, however, the enzyme with greater absolute activity in the leaves (Figure 2d) than in the roots was GS (Figure 2c). In the roots, however, the variations of GS-specific activity of M over NM plants, when comparing B (NM) and B (M) and A (NM) and A (M), were less than 40% and 35%, respectively.

In the leaves, the GS activity was markedly higher in the M than in the NM plants, where the percentage of activity increase in M compared NM plants in both locations was about 65%. This shows that in M plants, the specific activity of GS was greater in the leaves than in the roots, both in percentage increase and absolute value.

### 2.3. Free Amino Acids Content in NM and M Plants

Amino acid free content models showed significant differences in the roots (Figure 3a,c) and leaves (Figure 3b,d) when comparing the NM and M plants. At 190 DAS in the plants treated with mycorrhizae, the total free amino acid concentration increased to 70 and 100% in the roots and leaves of M, compared to the NM plants, for both A and B sites (Figure 3). The group of amino acids that showed a higher concentration in M than the NM roots includes glutamine (Gln), asparagine (Asn), alanine (Ala), arginine (Arg), ornithine (Orn) and citrulline (Cit). Although significant, smaller increases were observed for Glu and aspartate (Asp). The levels of leucine (Leu), valine (Val) and lysine (Lys) were over 100% higher in M than in NM roots. Glutamine, Arg, Asn and Cit, with high N/C ratios, are involved in N translocation and accumulation in roots. Large changes in concentration and distribution occurred also for the free amino acids in leaves (Figure 3b,d) of MS and NM plants at 190 DAS. The amino acids that contributed most to the total amount of N are Gln, Asn, Arg, Ala, Cit and Orn, which displayed the largest increase, reaching concentrations that were more than 200% greater, while Glu and Asp increased by about 75% in the leaves of M compared to NM plants in both sites (Figure 3b,d). In the leaves, however, only the Gln concentration was four times higher in the M compared to NM plants. Among the other amino acids, Leu, Val, Lys, Cit and Orn had concentrations higher than 200% also in the leaves of M plants relative to those of NM plants.

### 2.4. Isolation and Identification of Endophytic Fungi from Plant Roots in the Field

Unlike plant growth in a greenhouse or growth chamber, which takes place in more controlled conditions, experimentation in the field includes various pedoclimatic factors and microbiological factors in the soil that can significantly affect the physiological processes and growth response of plants. For the purpose of the present research, an investigation was conducted on the nature of the endophytic colonies hosted by the roots of the plants collected in the two sites in the field at each of which the same two different fertilization treatments (Section 4.2) were utilized.

Nine endophytic fungal taxa were isolated from the roots of 20 plants randomly distributed in the two investigated sites (Table 1). *Alternaria* and *Fusarium* were the two most common genera that were detected in the roots of many plant species. *Fusarium* species were observed in all the sites studied, while *Alternaria* species were mostly isolated in plant roots from site B. Besides those two genera, the fungi of the *Cladorrhinum macrophomina* species were detected in the samples examined by root isolation and ITS pyrosequencing with different frequencies. The fungi of *Cladorrhimum australe*, *Macrophomina phaseolina*, *Fusarium equiseti* and *F. avenaceum* species were isolated much more from the plant roots of site A (NM), with a percentage of frequency on 20 plants examined being 5%, 75%, 70% and 90%, respectively. The fungi of *Alternaria alternata*, *A. infectoria*, *F. equiseti* and *F*. *avenaceum* species were isolated much more from the plant roots of the A (M) site, with a percentage of frequency of 20 plants examined being 50% 55%, 40% and 55%, respectively. On the other hand, in the plant roots of the B (NM) site, the most significant attendance rates were in favor of *F. avenaceum*, *M. phaseolina*, *F. equiseti* and *F. algeriense* (85%, 50%, 50% and 40%, respectively) and in the plant roots of site B (M), in favor of *A. alternata*, *A. infectoria*, *F*. *avenaceum* and *M. phaseolina* (50%, 45%, 40% and 40%, respectively), as indicated in Table 1. and visibly evidentiable by the chromatism of the heat map Figure 4.

### 2.5. Cellulase Test on the Endophytic Fungi from Plant Roots

Cellulose, a linear polymer composed of β-d-glucopyranosyl units linked by β-1,4-d-glucosidic bonds, is a main component of plant cell walls. The cellulase test on isolated endophytes represents the response, using the cup plate method, of the ability of the fungi to lyse the cell wall of the host organism. Some of the strains isolated from the durum wheat roots in the two sites showed a positive test in the plate of the cellulase activity and, therefore, the ability of autonomous penetration into the root of the host plant (Table 2).

### 2.6. Root Microscopy

Although to different extents, all root samples examined showed endophyte colonization as observed via direct microscopy (Figure 5). In the seven images of Figure 5, it is possible to note also the presence of mycorrhizal colonization often associated with endophytic colonization in the same roots, highlighting vesicles and arbuscules.

The microsclerotia are the result of single or contiguous hyphae that swell and develop numerous septa. The subsequent lateral sprouting produces groups of spherical cells that subsequently become pigmented. The typical morphology of conidia and the conidial chains of *Alternaria* spp. From site B (NM) are clearly observed without the aid of histochemical reagents, using an optical microscope (Figure 6).

### 2.7. Evaluation of Fungal Competition via Dual Culture Assay of Alternaria tellustris

The DSE strain candidate *Alternaria tellustris* was isolated from healthy and mature roots of wheat plants and was evaluated for its fungal competition by using a dual culture assay. The results show that the DSE strain has an antagonistic activity against both the test pathogenic fungi *F. oxysporum* and *Rhizoctonia solani*. The degree of growth inhibition is 10.5% and 54.9% for *F. oxysporum* and 41.73% and 45.2 % for *R. solani* at 4 dpi and 8 dpi (days post-inoculation), respectively (Table 3, Figure 7).

## 3. Discussion

### 3.1. Root Mycorrhization and Plant Growth

Durum wheat seeds treated with a mixture of fungal spores *Glomus mosseae*, *G. intraradices* and *G. coronatum* caused the mycorrhization of the roots of durum wheat seedlings (M) in the open field where P availability was limited and N fertilization was reduced to 50% (i.e., 90 kg ha^−1^) compared to traditional treatments. The % RLC began to be significant only after 90 DAS at the beginning of stem elongation. Root samples obtained between 90 and 150 DAS showed hyperbolic growth in the percentage of mycorrhization, reaching a plateau of about 50% for both sites. In contrast, the percentage of spontaneous mycorrhization in roots that were nonmycorrhized (NM) does not exceed 10% at 100 DAS due to both the low microbiological load of the natural environment and the robust fertilization to which the non-mycorrhizal plants were treated. This reduction of mycorrhization is in accordance with the concept that the development of mycorrhizal colonization is affected by plant nutritional status and, more exactly, mycorrhization is reduced in response to the greater availability of macronutrients, in particular phosphorus and nitrogen, for the plant [37,38]. The mycorrhizal plants, compared to the non-mycorrhizal plants of the control, showed different growth responses: for both sites, the growth of mycorrhizal plants was initially slower, compared to control plants between 90 and about 130 DAS, suggesting that part of the energy resources of the plant is destined for the fungal colonization process, which slows down its growth. Subsequently, the plants treated with mycorrhizae showed an acceleration in the growth phase up to 150 DAS where the plants were strongly stimulated by the process of nitrogen assimilation, producing a significant increase in the amino acid pool that enhanced the overall, basic metabolism. These increases in metabolism were indirectly observed in the macroscopic evidence, with an increase in fresh weight of about 30%.

### 3.2. Nitrogen Assimilation

One of the most interesting, functional aspects of the symbiotic relationship between mycorrhizal fungi and the plant is the capacity of the fungus to assimilate inorganic nutrients from the soil and the subsequent transport of the mineral nutrients in the cortical parenchyma of the plant where metabolic photosynthates are exchanged. Arbuscular mycorrhizae can increase the absorption of N from the soil and its transfer to host plants [6,39]. Our results show that the activity of enzymes involved in nitrate assimilation, the reduction by NRA and the subsequent metabolism of ammonium by GS were highly correlated with root mycorrhization (Figure 2).

In site A at 130 DAS, when there was a high level of mycorrhization, M plants showed 55% and 40% higher NRA activity values compared to the NM plants for roots and leaves, respectively. In site B at 130 DAS during intense mycorrhization, M plants showed 80% and 55% higher NRA activity values compared to the NM plants for roots and leaves, respectively. Even with different values in mycorrhized plants there is a greater increase in NRA in the roots than in the leaves. Furthermore, at site B, this response was more pronounced: the increase was 45% and 37% more for site B than at site A for roots and leaves, respectively. In mycorrhizal maize roots, the improvement of the NRA activity has been assigned to the NADPH-dependent fungal NRA enzyme, which displayed activity four times greater with NADPH than with NADH [8]. This would suggest a hypothesis that the NRA activity, mainly due to the enzyme of the fungus, could be vastly higher. A confirmation of this comes from the fact that the fungus sends ammonium via the arbuscules to the cortical parenchyma where it is converted into glutamine [10]. However, nitrate reduction in the M plants was greater in both roots and leaves, indicating an interaction of mycorrhizal colonization with N nutrition.

Glutamine synthetase-specific activities (Figure 2) showed relatively small changes (<30%) in the roots for both sites, but GS activities increased significantly in the leaves of the plants treated with mycorrhizae (65% and 80%, compared to the NM plants for A and B sites, respectively). The high correlation between the percentage increase in NRA and GS activities of roots and leaves tends to generate a spatial separation between the assimilative reduction of nitrate (mostly in the roots) and ammonium metabolism (mostly in the leaves). Moreover, there is kinetic evidence that the chloroplast-localized GS isoform (GS2) of wheat (*Triticum aestivum* L. cv. Jubilejnaja-50) takes place at the carbon–nitrogen metabolic branch point, and that its enzymatic activity is regulated also by the availability of C, whose main source in photosynthesis is the leaves [40].

### 3.3. Variation of Free Amino Acid Level and Distribution in Roots and Leaves of Mycorrhized Durum Wheat Plants

The amino acid patterns of mycorrhizal roots and leaves (Figure 3) include elevated levels of Gln and the low levels of Glu, further confirming that the assimilation of ammonium through the GS-GOGAT pathway occurred both in the plant and in the fungus [9,41].

The elevated concentrations of Arg and Orn strongly support the current model of conversion of nitrogenous forms in AM symbiosis, which includes the synthesis of Arg in the extra-radical mycelium and the transfer of Arg to the intraradical mycelium, where it is broken down by arginase and urease, and from whence it is subsequently transferred to the root cells as ammonium [10]. The elevated levels of Arg and the possibility that it can be transferred partly to the root cells demonstrate the increase in Cit derived from Orn through the Ornithine transcarbamylase and Carbamoyl phosphate synthetase by Gln.

Given the the high N/C ratio, we see that Cit predominates in transported N while, in some other instances, Arg is the major N-solute of xylem [42]. The decrease in Asp supports its role in the synthesis of argininosuccinate, an intermediate of Arg synthesis [43]. An inevitable increase in oxygen demand in the roots that developed from the seeds treated with mycorrhizae could trigger in the cells of the cortex a metabolic response, with low oxygen content, through the down-regulation of oxygen consumption pathways, such as respiration and enrichment in organic reserves [44,45]. The elevated levels of Ala in the roots treated with mycorrhizae suggest that glycolysis, more than overall respiration, was increased to support plants and fungi to form and maintain the high turnover rate of arbuscules [46,47]. In *Medicago truncatula*, a cytological change corresponding to an increased need for plastid and mitochondrial products during the establishment and functioning of symbiosis was observed. The mitochondria were thickened around the arbuscule, and this explains the change in the plastid ultrastructure caused by a reduction in carbohydrate reserves [45]. Finally, the Glu in the leaves reached, under various conditions, concentrations of 80–150% greater than in the roots, and together with the higher levels of Gln (Figure 3) and the activity of GS (Figure 2), it is probable that GOGAT enzymes were highly functional in the synthesis of nitrogen compounds from carbon and energy skeletons, supplied by photosynthesis.

### 3.4. Fungal Screening in Roots of Wheat Plants and Possible Antagonism between Mycorrhizal Fungi and Pathogenic Endophytes

An analysis to explore indigenous endophytic fungi in mycorrhized and non-mycorrhized durum wheat roots enables one to recognize indicators of healthiness and pathogenicity and reveals the potential ability of endophytes to colonize roots in competitive relationships with mycorrhizal fungi in soil intended for agricultural crops. At both sites, *F. avenaceum* and *F. equi*seti are the fungi with the highest rate of frequency in the NM roots with average values of 85% and 60%, respectively. They can be considered to be the most common species of *Fusarium* on cereals in temperate climates, and Fusarium fungi are generalist pathogens responsible for diseases in numerous crop species. *Fusarium* spp. produce mycotoxins, which are factors of pathogenicity and virulence in various plant–pathogen interactions. *Macrophomine phaseoline* was also found in the NM roots of both sites with significant frequencies and average values of approximately 60%. This necrotrophic fungus is an important pathogen of many crops, such as corn, sorghum, potato and soy.

The fungal screening performed on samples from the two sites indicated an interesting result that suggests the potential influence of mycorrhization on the fungal population. The presence of *F. avenaceum*, *F. equiseti* and *M. phaseoline* was strongly reduced by mycorrhization at both sites.

### 3.5. Spontaneous Colonization of Soil DSE, an Ecological Niche between Saprophytic Microorganisms and Pathogens of the Root System

Unlike the experiments conducted in growth chambers, in which the development of mycorrhizae, or mycorrhization, occurs in roots growing in relatively sterile soil, the experimental conditions in an open field may include the presence and possible conditioning of the endophytic fungi affecting the symbiotic action of mycorrhizae.

The molecular identification of isolated endophytic strains from durum wheat roots shows their presence in all the samples examined, including roots treated with mycorrhizae, suggesting the autonomous ability of mycorrhizae to colonize the root. This hypothesis of endophytic colonization was confirmed with the positive activity of mycorrhizae being demonstrated in the cellulase test, indicating that a group of hydrolytic enzymes was capable of hydrolyzing the organic polymer cellulose to smaller sugar components.

Among the isolated strains, fungi of the typically pathogenic genus Fusarium were found at both sites, but with different frequencies in the various root samples examined, depending on the site of origin of plants that already established mycorrhizal symbiosis. The other endophytic fungi present in the root system, such as *A. alternate*, also show different frequencies of distribution among the samples examined. These organisms, exerting an interaction with host plants, seem to play a wide range of ecological roles, constituting a continuum from mutualism to parasitism [48,49,50]. Particular interest has been shown in recent years regarding endophytic fungi considered to be dark septate endophytes (DSE), a group of endophytic fungi characterized by their morphology of melanized, septate hyphae. DSE are frequently observed in interaction with mycorrhizal fungi on shrubs, such as ericoid plants, and as ectomycorrhizae on orchid plants [51]. The effects of DSE on host plants range from pathogenic to mutualistic, depending on environmental factors and the genotypes of hosts and fungi [27]. The frequency of distribution of DSE in the cortical cells of the samples examined in the present report would confirm the tendency of DSE to colonize roots that are already mycorrhizal, with a percentage ranging from 20% to 40% more than in non-mycorrhizal roots. The greater presence of DSE coupled with mycorrhizal associations reduces the vital space for the development and colonization of pathogenic endophytes, including *Fusarium*, as confirmed by the low percentages of frequency in our mycorrhizal samples. On the other hand, some DSE, among which is *A. alternata*, have a wide range of metabolites that possess a great variety of biological activities, such as antimicrobial functions and antioxidant properties [52]. These DSE can effectively counteract, through biocontrol, the root contaminations of bacteria and fungi that are pathogenic to the plant. Some pathogenic fungi or bacteria that enter the xylem vessels that are the primary conduits for water and minerals of a vascular plant can proliferate within the vessels, causing the xylem flow to be blocked. This blockage can occur when fungi such as *F. oxysporum* gain entry to xylem vessels [53]. The ambiguous behavior of the genus *Alternaria* from leaf spot pathogen to root saprophyte can be attributed to the different conditions and availability of oxygen, which, at low levels in the roots, would reduce the virulence of the fungus by radically modifying the biotrophic response [54]. This hypothesis is visibly supported by the image in Figure 5D, where the colonization of cortical cells by *A. infectoria* (site B, M) does not generate cellular necrosis or damage. In fact, DSE produce microsclerotia and hyaline hyphae in the roots [55]. Moreover, it is possible to hypothesize, on the basis of the image (Figure 5D) that when microsclerotia colonized the epidermis and of the radical cortex, hyphae became connected inter- and intracellularly to the external mycelium. This type of cytological interaction between root and DSE facilitates the translocation of nutrients from soil to cortical cells through a network of fungal hyphae laying the foundation for an incipient symbiotic relationship between DSE and root system.

The capabilities of DSE to diminish the availability of nutrients and space to pathogenic fungi and also to promote plant defenses and plant growth [55] increases the interest in their role in the biological control of plant diseases. From our isolates among non-pathogenic DSE, plugs of mycelium of *A. tellustris* were used in an antimicrobial dual culture assay test to determine whether a DSE is able to counteract terrestrial plant pathogenic fungi, such as *F. oxysporum* and *R. solani*. The analyses show that *A. tellustris* can diminish the nutrients and space available to both terrestrial pathogens. These results appear to indicate that DSE have an important role in diminishing the nutritional and spatial resources available to terrestrial plant pathogens, using different mechanisms of action, as mentioned by Santos et al. [56]. The present research suggests that endophytic fungi, such as *A. tellustris*, may have a role in the induction of a plant’s defensive responses, such as systemic acquired resistance (SAR) and induced systemic resistance (ISR). The SAR response is caused by the activation of genes encoding several pathogenesis-related proteins (PRs) through the salicylic acid (SA) biosynthetic pathways. The induction of ISR is caused by upregulating several genes involved in the jasmonic acid (JA) and ethylene (ET) pathways. Lahlali et al. [57], for example, in one of the first reports on the molecular mechanisms of induced resistance mediated by a non-mycorrhizal endophytic fungus, report that the DSE *Heteroconium chaetospira* strain BC2HB1 is able to suppress the clubroot on canola through induced resistance via the concerted upregulation of genes involved in JA, ET and auxin biosynthesis. The PR-2 protein may also be involved in the plant’s defenses [55].

## 4. Materials and Methods

### 4.1. Mycorrhizal Spore Inoculum for Seeds

Studies on the physiological effects of the mycorrhization of *durum wheat* (cv. Iride) were conducted in 2019 at two different sites within the same pedoclimatic area near Termoli Italy: Larino 700 m a. s. l. (A); Rotello 640 m a. s. l. (B). The mycorrhizal inoculum used contained a mixture of symbiotic fungi (*Glomus mosseae*, *G. intrardices* and *G. coronatum* 0.0001% *w*/*w*) provided by MS Biotech S.p.A. (Larino SS 87, km 204) and was sprayed with a backpack pump on about 450 kg of *durum wheat* seeds before sowing.

### 4.2. Physical and Chemical Analysis of the Soil Samples

Physical and chemical analyses of the soil were performed in the laboratory in accordance with official Italian procedures [58]. Coarse sand (2.0–0.2 mm), fine sand (0.2–0.05 mm), silt (0.05–0.002 mm) and clay (<0.002 mm) fractions were separated via pipette and wet sieving following pre-treatment with H_2_O_2_ and sodium hexametaphosphate. Textural classes were assigned according to Italian Soil Science Society standards [58]. Soil pH was measured potentiometrically in soil solution suspensions of 1:2.5 H_2_O. Total CaCO_3_ (%) was determined using a De Astis calcimeter. Total organic carbon (TOC) was determined via wet digestion through a Walkley–Black procedure. Total nitrogen (Ntot) was determined using the Kjeldahl method. Phosphorus was determined using Olsen’s method. Cation exchange capacity (CEC) and exchangeable-base cations were extracted with 0.5 N BaCl2-TEA at pH 8.2 and determined by atomic absorptions spectrophotometry. The data were obtained by performing three replicas, and the mean ± SD was thus calculated (Table 4).

### 4.3. Fertilization Plan

The cultivable area of each of the two sites was 0.25 ha, divided into two symmetrical parcels, one intended for the non-mycorrhized sowing (NM) of parcel 1, the other for the mycorrhized sowing (M) of parcel 2.

On the basis of the preliminary soil analyzes, 3 fertilizations were conducted: the first was background fertilization, the second was during the phenological phase of tillering and the third was during the phenological phase of the beginning of stem elongation. In the parcels cultivated with seeds and mycorrhizal spores, no basic fertilization was performed and no phosphorus fertilizer was added (Table 5).
(1)Basic fertilization: 100 kg per hectare (kg ha^−1^)Fertilizer composition: 6% total nitrogen (1% organic N; 2% NO_3_^−^-N; 3% ureic N);18% total phosphoric anhydride (P_2_O_5_); 32% water-soluble P_2_O_5_; 20% total CaO; 15% water-soluble sulfur anhydride (SO_3_); 7.5% organic carbon of biological origin.(2)Phenological phase of the beginning of tillering: nitrogen fertilizer 180 kg ha^−1^ (NM) parcel 1 and 90 kg ha^−1^ (M) parcel 2.Composition: 30% ureic N; 30% NO_3_^−^-N; 15% water-soluble SO_3_. Fertilization was repeated after a week.(3)Phenological phase of the beginning of stem elongation: nitrogen fertilizer 200 kg ha^−1^ (NM) parcel 100 ha^−1^ (M) parcel 2. Composition: 30% ureic N; 30% NO_3_^−^-N; 20% water-soluble SO_3_. Fertilization was repeated after a week.

### 4.4. Sampling, Plant Growth and Mycorrhizal Colonization Analyses

Samples were taken 90 (beginning of tillering stage), 110 (beginning of stem-elongation stage), 130, 150, 170 and 190 days after sowing (DAS) following the treatment of durum wheat seed with mycorrhizae. At each sampling, ten plants for each parcel of experimental land were collected at ten randomly distributed points, and were used for analyses.

The roots were cleaned in a container with water and then with sterile water, then dried with a paper towel and promptly used to determine the percentage of mycorrhization using the methods of Giovannetti and Mousse [59]. The shoots were used to calculate root fresh weight during the growth phase of the plants. Each subsample was counted three times by reorganizing the roots in the Petri dish, and the mean ± SD was calculated.

### 4.5. Primary Amino Acid Analyses

Proline was determined using high-performance liquid chromatography (HPLC) with a fluorescent 9-fluorenylmethoxycarbonyl derivative (P-FMOCcarbamate) using aliquots of root and leaf extracts that were formerly derivatized using o-Phthaldialdehyde Reagent Solution (OPA) solution to exclude the primary amino acids and was fluorometrically detected by excitation at 266 nm and emission at 305 nm [60].

Protein concentrations in root and leaf samples were determined using the methods of Lowry et al. [61].The values are means ± SD of ten plants (*n* = 10). Values are statistically different at *p* ≤ 5% according to Tukey’s test.

### 4.6. Nitrate Reductase Assimilatory and Glutamine Synthetase Activity Assays

Portions of 300 mg FW and 1 g FW of leaf and root, respectively, after being immersed in liquid nitrogen, were finely ground in an agate mortar and subsequently extracted in 10 mL extraction buffer, and NRA and GS activities were determined using the methods of Gibon et al. [62]. The values are means ± SD of ten biological replicates (*n* = 10). Values are statistically different at *p* ≤ 5% according to Tukey’s test.

### 4.7. Root Microscopy

Direct observation of natural host root colonization by mycorrhizal and endophytic fungi was conducted using light microscopy as in Bordallo et al. [63]. Six to eight root pieces (c. 0.5 cm long) per plant sample were cut from each root system. Arbuscular mycorrhizal tissue was analyzed by evaluating the presence of aseptate hyphae, hyphal coils, arbuscules or arbusculate coils with or without vesicles. Root sections were then treated with Trypan blue, as previously indicated. The excess stain was removed with distilled water and the sample was blotted onto a filter paper. Root samples on microscope slides were observed and photographed with an Olympus BH-2 microscope.

### 4.8. Isolation of Endophytic Fungi

Regarding dark septate endophytes, fungal colonization was characterized using regularly septate, melanized or hyaline hyphae with microsclerotia or moniliform cells that allowed us to discriminate the endophytic structures from the mycorrhizal structures.

Endophytic fungi were isolated from fresh plant roots that were disinfected by soaking in 75% ethanol (0.5 min), 3% sodium hypochlorite (5–8 min) and 75 % ethanol (1 min), and which were then rinsed twice for 1 min in sterile water. The plant roots were cut into small pieces (0.5 cm) using a sterilized scalpel. Segments from each plant tissue sample were randomly chosen and placed in Petri dishes containing potato dextrose agar (PDA) supplemented with 50 µg mL^−1^ streptomycin and 50 µg mL^−1^ chloramphenicol. These plates were incubated at 28 °C until fungal growth appeared. The colonies were counted and grouped by their morphologic characteristics, and representative isolates of fungal diversity were collected, purified and preserved for future analysis. All the materials used were purchased from Merck KGaA, Darmstadt, Germany.

### 4.9. Molecular Identification of Fungi

Due to the difficulty of identifying the morphological characteristics of endophytes isolated from the durum wheat roots, the identity of the groups of fungal isolates was confirmed by means of molecular methods.

Fourteen hypothetical endophytic fungi were analyzed. The isolated strains were aerobically cultured in 250 mL Erlenmeyer flasks containing PDA broth at 28 °C for 5 days without shaking. Each isolate (1 mL) was centrifuged at 14,000 rpm for 5 min at 4 °C, and the pellet obtained was subjected to DNA extraction using a DNA extraction kit (Macherey-Nagel, NucleoSpin Tissue Kit, Duren, Germany) according to the manufacturer’s instructions. The quantity and purity of the DNA were assessed using the NanoDrop spectrophotometer 2000 (Thermo Scientific, Wilmington, DE, USA).

The amplification of internal transcribed spacer (ITS) region of rDNA was carried out using universal primers ITS1 (5′-TCCGTAGGTGAACCTGCGG-3′) and ITS’ (5′-TCCTCCGCTTATTGATATC-3′). Polymerase chain reaction (PCR) determination was performed using a Mastercycler nexus gradient (Eppendorf, Hamburg, Germany). The reaction mixture consisted of 10 µL TaqMan 2x PCR Master Mix (Norgen Biotek Corp., Thorold, ON, Canada), 2 µL of each primer (2.5 µM), 2 µL template DNA and the nuclease-free water were added to bring the final volume to 20 µL.

PCR products were separated in 1.5% (*w*/*v*) agarose gel via electrophoresis for 45 min at 80 V in 1X TAE buffer (Thermo Fisher Scientific, Waltham, MA, USA) and were subsequently visualized using a UV transilluminator (Bio-Rad molecular image Gel Doc XR, Hercules, CA, USA) after ethidium bromide (50 µg/mL, Merck KGaA, Darmstadt, Germany) staining. After purification (QIA-quick PCR Purification kit, QIAGEN GmbH, Hilden, Germany), the DNA products were sent to a commercial facility for sequencing (Macrogen Europe BV, Meibergdreef, Amsterdam, The Netherlands). The Basic Local Alignment Search Tool Nucleotide (BLASTN) tool encoded in NCBI suite [64] was applied to the GenBank [65] database to identify the sequences.

### 4.10. Evaluation of Fungal Competition by Dual Culture Assay of Alternaria tellustris

*Alternaria tellustris* was used as a non-pathogenic DSE candidate to evaluate the potential antimicrobial activity of this strain against pathogenic fungi of terrestrial plants. The strain was screened for its antagonistic activity via the double culture method against two phytopathogenic fungi: (i) *Fusarium oxysporum* f.sp. *lycopersici* and (ii) *Rhizhoctonia solani*. The plugs of mycelium (6 mm Ø) were taken under aseptic conditions every 4 to 5 days, selecting the pure cultures of *Alternaria tellustris* and each phytopathogenic fungus, and were then transferred to a Petri dish (90 mm Ø) containing 20 mL of potato dextrose agar (PDA) and kept 6 cm separate from each other. The plates were incubated at 28 °C for 8 days and the treatments were performed in triplicate. Pathogen and endophyte growth was observed daily, and radial growth was recorded by measuring the mean colony radius on day 4 and day 8 after inoculation (dpi). As a control, each fungus was plated by itself. The percentage of inhibition of the tested DSE and phytopathogenic fungi was calculated using the formula:% inhibition = (R1 − R2/R1) × 100
where R1 is the radial growth of the control plate, and R2 represents the radial growth of the dual culture plate. The data were obtained by performing three replicas and the mean and ±SD was then calculated.

### 4.11. Statistical Analysis

Data are presented as means ± SD of the 10- or 20-plant samples randomly distributed in the experimental parcel. Each analysis of each sample was replicated two times. The statistical significance of differences was calculated at using Tukey’s test (*p* ≤ 5%). An analysis of variance was performed using the software IBM_ SPSS_ Statistics version 22.0 (SPSS Inc. Chicago, IL, USA, 2014).

## 5. Conclusions

Mycorrhized wheat plants showed a more efficient assimilation of inorganic nitrogen than did control plants at two sites in open fields. The increase in specific NRA and GS activities and the consequent enrichment of the amino acid pattern resulted in an increase in plant growth and yield under conditions of low fertilization and minimal environmental impact. The spontaneous colonization of the roots of durum wheat by endophytic fungi, such as DSE that were present in the soil, appears to be favored by the presence in the root of mycorrhizal fungi. The consociated interaction of DSE and mycorrhizal fungi appears to have been established, resulting in increased N absorption by the durum wheat. This relatonship between DSE, mycorrhizal fungi and the roots resulted in (1) the improved the growth of the plant, (2) increased competitiveness between symbiotic fungi and pathogenic fungi, (3) reduced colonization spaces and (4) reduced pathogenic contamination, based upon measurements showing diminished presence of *Fusarium* species at two cultivation sites. Ultimately, it is possible to agree that the physiological role of these endophytic fungi could be relegated to the prevention of pathogenic infections and to the transport of nutrients from the soil to the roots in parallel with mycorrhizal associations.

## Figures and Tables

**Figure 1 plants-11-00924-f001:**
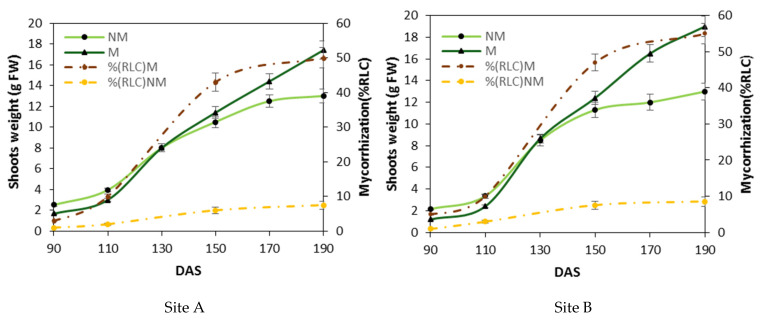
Differences in the aerial biomass weight between mycorrhized and non-mycorrhized plants. Shoot weights (a mean of ten shoots were in each parcel of control (NM) and mycorrhized plants (M)) and the corresponding root length colonization (% RLC) between 90 and 190 days after sowing (DAS) at two separate locations (Larino, Site A; Rotello, Site B). The mycorrhization index (dashed line) is expressed as a percentage of root length colonization % RLC by mycorrhizae. Bars indicate the standard deviation (±SD), and the means are the mean values of ten plants (*n* = 10) randomly distributed in the experimental parcel.

**Figure 2 plants-11-00924-f002:**
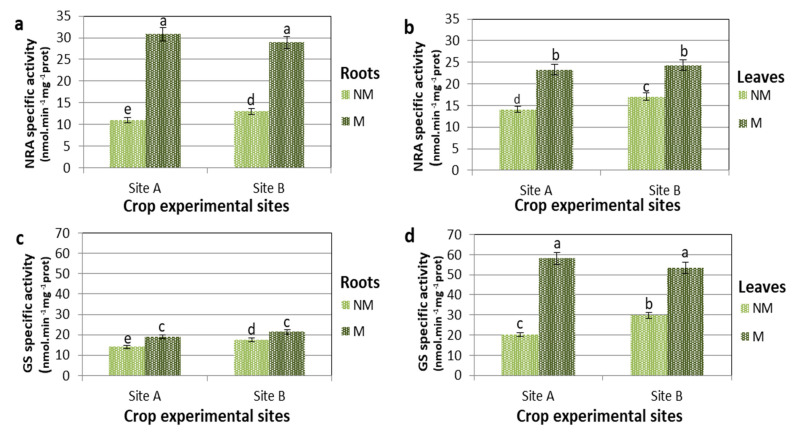
Nitrate reductase (NRA) and Glutamine synthetase (GS)specific activities of control (NM) and M plants *of Triticum durum* grown in two different soil sites (A and B) and collected at 190 DAS. Data are for NRA in roots (**a**) and in leaves (**b**), and GS in roots (**c**) and leaves (**d**). Specific activities are expressed in nmol min^−1^ mg^−1^ protein. The values are means ± SD of ten biological replicates (*n* = 10). Values marked by common letters are not statistically different at *p* ≤ 5% according to Tukey’s test.

**Figure 3 plants-11-00924-f003:**
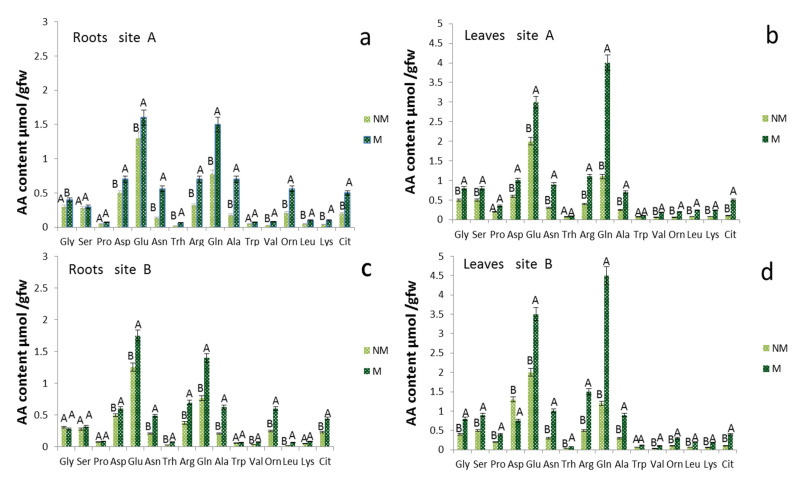
Amino acid concentrations of roots and leaves tissue of *Triticum durum* of control (NM) and mycorrhizal (M) treatments 190 d after sowing in two different soil sites (A and B; (**a**,**c**) refer to the roots, while (**b**,**d**) to the leaves). The values are means ± SD of ten plants (*n* = 10) in each experimental parcel. Bars indicate the standard deviation. Values marked by common letters are not statistically different at *p* ≤ 5% according to Tukey’s test.

**Figure 4 plants-11-00924-f004:**
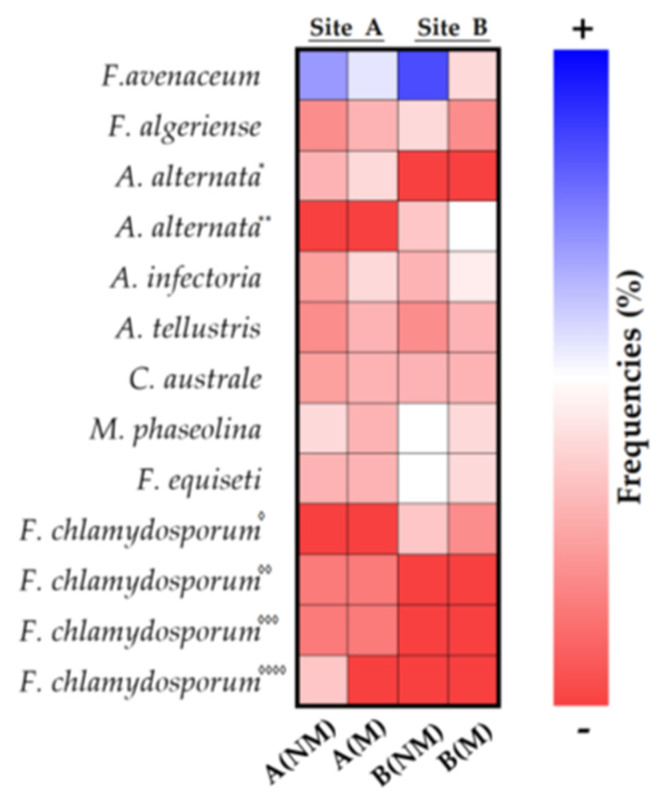
Heatmap of percentage frequency of endophytic fungi in the roots of *Triticum durum*. Different strains are indicated by superscript asterisks (“*”, “**”) and diamonds (“^◊^”, “^◊◊^”, “^◊◊◊^” and “^◊◊◊◊^”).

**Figure 5 plants-11-00924-f005:**
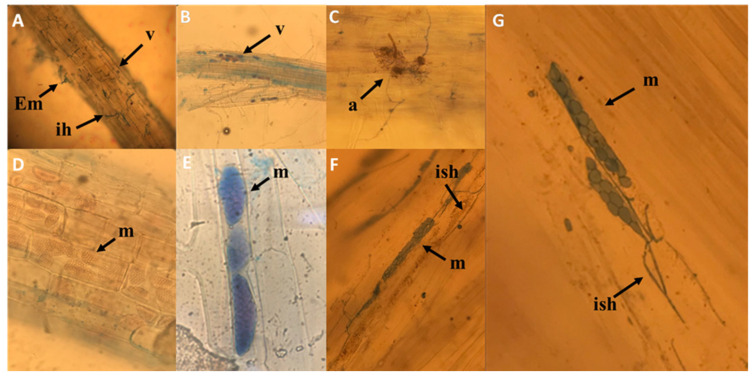
Light microscopy of root colonization by mycorrhizal fungi and DSE. Vesicles and arbuscules (**A**–**C**); intracellular growth of endophytic fungus with stained microsclerotia at the young stage (arrowhead) in cortical cells without and with trypan blue (**D**–**G**). Em= external mycelium; ih = intracellular hyphae; v = vesicles; a = arbuscule; m = microsclerotia; ish = intracellular septate hyphae.

**Figure 6 plants-11-00924-f006:**
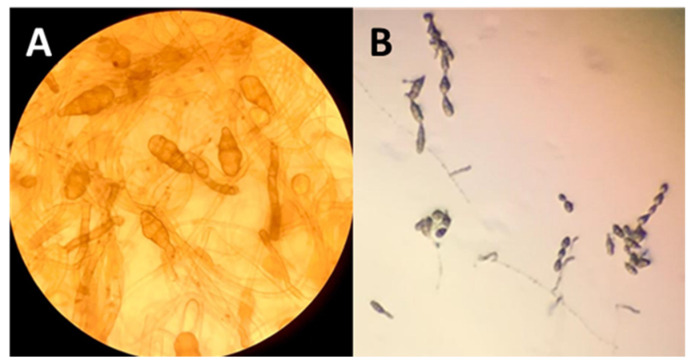
Morphology of *Alternaria* spp. isolated from roots. (**A**) Conidia and (**B**) conidial chains of *Alternaria* spp.

**Figure 7 plants-11-00924-f007:**
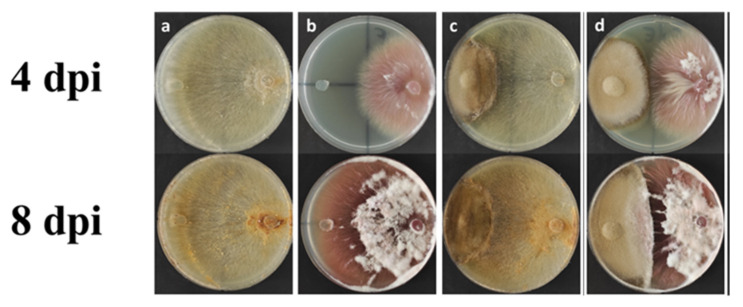
Representative pictures of the antagonistic activity assay via dual culture method at 4 and 8 days post-inoculation (dpi), respectively. (**a**,**b**) Control plates of *R. solani* and *F. oxysporum*, respectively; (**c**) dual culture assay of *A. telluris* and *R. solani*; (**d**) dual culture assay of *A. telluris* and *F. oxysporum*.

**Table 1 plants-11-00924-t001:** Percentage frequency of endophytic fungi isolated from the roots of 20 *Triticum durum* plants harvested at sites A and B not treated with mycorrhizae NM and treated with mycorrhizae M. Different strains are indicated by superscript asterisks (“*”, “**”) and diamonds (“^◊^”, “^◊◊^”, “^◊◊◊^” and “^◊◊◊◊^”).

Microorganisms in Roots	Gene Bank Code	% Frequency Site A		% Frequency Site B	
		NM	M	NM	M
*Fusarium avenaceum*	OM417228	90%	55%	85%	40%
*Fusarium algeriense*	OM417243	20%	30%	40%	20%
*Alternaria alternata* *	OM422688	30%	50%	-	-
*Alternaria alternata* **	OM422689	-	-	35%	50%
*Alternaria infectoria*	OM422690	25%	55%	30%	45%
*Alternaria tellustris*	OM422702	20%	30%	20%	30%
*Cladorrhinum australe*	OM422711	65%	30%	30%	30%
*Macrophomina phaseolina*	OM422725	75%	30%	50%	40%
*Fusarium equiseti*	OM422743	70%	40%	50%	35%
*Fusarium chlamydosporum* ^◊^	OM422742	-	-	35%	20%
*Fusarium chlamydosporum* ^◊◊^	OM429358	15%	-	-	-
*Fusarium chlamydosporum* ^◊◊◊^	OM429359	15%	-	-	-
*Fusarium chlamydosporum* ^◊◊◊◊^	OM429357	35%	-	-	-

**Table 2 plants-11-00924-t002:** Result of cellulase test on isolated endophytic strains in roots of *Triticum durum.*

Isolates	Results
*Fusarium avenaceum*	+ +
*Fusarium algeriense*	+ +
*Alternaria alternata* *^/^**	+ + +
*Alternaria infectoria*	+ + +
*Alternaria tellustris*	+ +
*Cladorrhinum australe*	+
*Macrophomina phaseolina*	–
*Fusarium equiseti*	+ +
*Fusarium chlamydosporum* ^◊^ ^/^ ^◊◊◊◊^	+ +

The + symbol corresponds to 0.5 cm radius of the cellulase activity halo. The minus (−) symbol corresponds to no-halo. *^/^** (one / two); ^◊/◊◊◊◊^ (one / four).

**Table 3 plants-11-00924-t003:** Percentage of growth inhibition by *A. tellustris* against *F. oxysporum* and *R. solani*. Data show the mean ± SD of three replicas (*n* = 3) at 4 and 8 days post-inoculation.

Days Post-Inoculation	% of Inhibition by *A. tellustris*	
	*Fusarium*	*Rhizoctonia*
4	10.5 ± 2.0	41.7 ± 1.1
8	54.9 ± 2.7	45.2 ± 0.8

**Table 4 plants-11-00924-t004:** Physico-chemical determinations of the soils of sites A and B.

Larino (A)		Rotello (B)	
coarse sand	18% ± 3%	coarse sand	11% ± 2%
fine sand	24% ± 4%	fine sand	36% ± 6%
silt	32% ± 6%	silt	13% ± 3%
clay	38% ± 7%	clay	40% ± 7%
pH	7.36 ± 0.2	pH	7.4 ± 0.1
CaCO_3_ total	9% ± 1.5%	CaCO_3_ total	9% ± 1.8%
CEC	18 cmol (+) kg^−1^ ± 2	CEC	19 cmol (+) kg^−1^ ± 2
total organic carbon	8.4 g kg^−1^ ± 1.5	total organic carbon	8.5 g kg^−1^ ± 1.5
total Nitrogen	1.1 g kg^−1^ ± 0.2	total nitrogen	1.2 g kg^−1^ ± 0.2
P_2_O_5_	19.2 mg kg^−1^ ± 3	P_2_O_5_	19.7 mg kg^−1^ ± 3
C/N	7.6 ± 1.5	C/N	7.1 ± 1.4

CEC: Cation exchange capacity. Data show the mean ± SD of three soil sample replicas (*n* = 3).

**Table 5 plants-11-00924-t005:** Fertilization plan.

Treatment	Date	Phenological Stage	A (NM)	A (M)	B (NM)	B (M)
Basic fertilization	2019/10/10	-	100 kg/ha^−1^	-	100 kg/ha^−1^	-
2nd fertilization	2020/02/04	Tillering	90 kg/ha^−1^	45 kg/ha^−1^	90 kg/ha^−1^	45 kg/ha^−1^
2nd fertilization	2020/02/18	Tillering	90 kg/ha^−1^	45 kg/ha^−1^	90 kg/ha^−1^	45 kg/ha^−1^
3rd fertilization	2020/03/14	Stem extension	100 kg/ha^−1^	50 kg/ha^−1^	100 kg/ha^−1^	50 kg/ha^−1^
3rd fertilization	2020/03/26	Stem extension	100 kg/ha^−1^	50 kg/ha^−1^	100 kg/ha^−1^	50 kg/ha^−1^

## Data Availability

The data can be found and made available by the laboratory archive of Plant Physiology of the University of Molise.

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
