# Peer review of "Mycorrhized Wheat Plants and Nitrogen Assimilation in Coexistence and Antagonism with Spontaneous Colonization of Pathogenic and Saprophytic Fungi in a Soil of Low Fertility"

_plants, 2022, doi:10.3390/plants11070924_

Round 1

Reviewer 1 Report

General comment:

The manuscript aimed to investigate the interactions between endophytes and mycorrhizal durum wheat roots and their effects on productivity in two fields with low P and N fertilization rates were used; endophytic fungi were also isolated and identified. Biochemical and molecular tools were properly used to get a set of data statistically evaluated and clearly presented and discussed.

The topic is interesting and fits the aims of the journal, the manuscript is quite well written with minor language and editing errors, below reported, the overall evaluation is positive and the suggestion is to accept it for publication with minor changes.

Specific remarks:

l.64-72: quite generic statement; please reword or delete!

Fig1.: no statistics?

l.162-171: please move this statement to the discussion section.

Fig.3: why fw and not dw? Please justify or modify.

l.417: …was confirmed “by” their positive activity…. (add “by”).

l.677: change to with two.

l.703-710: acknowledgments and conflicts of interests are repeated twice!

Author Response

Reviewer 1

General comment:

The manuscript aimed to investigate the interactions between endophytes and mycorrhizal durum wheat roots and their effects on productivity in two fields with low P and N fertilization rates were used; endophytic fungi were also isolated and identified. Biochemical and molecular tools were properly used to get a set of data statistically evaluated and clearly presented and discussed.

The topic is interesting and fits the aims of the journal, the manuscript is quite well written with minor language and editing errors, below reported, the overall evaluation is positive and the suggestion is to accept it for publication with minor changes.

Specific remarks:

l.64-72: quite generic statement; please reword or delete!

The authors agree, but even if the statement may appear generic, the terms “positive”, “neutral” and “negative”, are part of the technical language of ecomicrobiology and ecobotany. (see: Kefi et al. 2016 Mechanism and consequences of facilitation in plant communities.When can positive interactions cause alternative stable states in ecosystems? Functional Ecology, 30, 88–97

However, in response to Reviewer 1’s comment, we replaced the terms with “beneficial”, “harmless” and “harmful”.  The statements expressed have been taken from the current literature. See also: Nelson 2019  Mutualism, Commensalism, Parasitism.  Trend science

Fig1.: no statistics?

The statistics have now been providedc.

l.162-171: please move this statement to the discussion section.

You’re absolutely right! Thanks.

The change suggested by Reviewer 1 has been made.

Fig.3: why fw and not dw? Please justify or modify.

Certainly. In the department we have hot air dryers, but we preferred to refer to fresh weights for technical reasons. Unlike animal tissues, tissues of growing plants can contain more than 90% water. Therefore, for small leaf samples the dry weight is reduced to a few tens of mg with the risk of a greater incidence of the technical weighing error. That’s why we normalize everything to fresh weight. Problems of weighing accuracy can alsoarises when the plant is not fully hydrated as water or saline stress, but those problems did not occur under the conditions of our investigation..

l.417: …was confirmed “by” their positive activity…. (add “by”).

This suggestion is now part of the text.

l.677: change to with two.

This has been corrected.

l.703-710: acknowledgments and conflicts of interests are repeated twice!

Thank you! The redundancy has been eliminated.

Reviewer 2 Report

I was very positive towards this manuscript, but it is rather difficult to catch the real meaning of this study and relevance of the obtained results behind what was written. The experimental setup cannot be fully understood and the results properly evaluated. I have an opinion that incomprehensiveness results mostly from poor communication, not from poor and unsound experimentation. But I may be wrong in this respect. There are so many difficult to understand points, that it is even impossible to indicate all of the discrepancies and problems. I will try to raise the main points though except discussion, which is needless without understanding the experimental part.

Title

Too long and complicated. It is necessary to concentration on the main aspects of the study, related to mycorrhiza/endophytic and pathogenic fungi interaction. Do not introduce abbreviations. Do not use redundant capitalization ("dark septate").

Abstract

Not well balanced, difficult to understand nature of performed experiment and the main findings. Do not use unintroduced abbreviations (DSE) and do not introduce abbreviations (NR, GS), as these will not been used further in the Abstract. Do not use redundant capitalization as for enzyme names. Explain what type of experiment was performed (field study, followed by some laboratory tests?), what was modified/manipulated during the experiment. Use past tense exclusively when describing results.

– line 26–27, is it meant that "nutrient supply from mycorrhizal fungi to plant root tissues resulted in increased plant growth"?

– line 30, what is meant by "endophytic fungal radicals"?

Introduction

Starts rather abruptly, as general part is missing, pointing to particular field (practical or fundamental) to which this study is related. Further, the main logical thread of analysis seems to be missing, as the Introduction consists of separate statements without logical connection between them.

Not sure if "cytological interactions" is the best term to describe physiological and biochemical complex of interaction occurring on plant/mycorrhiza interface and affecting plant functions and growth (line 41).

References are completely missing in the 4th paragraph (lines 61–75). Each fact needs to be supported by a reference. Information included is very general, textbook level text. "Trichoderma" (line 62) is misspelled.

Do not use redundant capitalization (dark septate endophytes) (line 76).

The last paragraph is rather incomprehensible, the authors need to sort out the information and to concentration on the aspects leading to the aim of the study.

The aim needs to be reformulated, as it seems that the real emphasis in this study is on fungal interactions, not functional aspects of nitrogen metabolism, and that these two aspects have been combined "by force" not by functional relatedness. May be the main aim initially was towards physiological aspects of nitrogen assimilation in wheat plants due to mycorrhiza, but fungal interactions just appeared during the study and therefore were included?

Materials and methods

In the very beginning, it needs to be clearly stated what kind of field experiments as well as laboratory tests were performed to understand rationale/necessity/complexity of the whole study. At present, it cannot be understood if there were treatments with/without mycorrhizal inoculation vs different fertilizer levels.

Line 492 and further, quintal is not a scientifically legitimate unit of measurement, use SI system units only. Plant scientific name needs to be indicated. Use "cultivar" (cv) instead of "variety" (var.). Indicate that only part of seed material was inoculated with mycorrhizal fungi.

Do not use redundant capitalization (total organic carbon, cation exchange capacity).

Line 505, information is incomplete, CEC cannot be "extracted".

Line 506, explain what is TEA.

"4.2. Fertilization plan" is very poorly written. Better provide a scheme in a form of a table how different treatments were performed, including both phenological phase and exact timing of treatment. Do not divide this in subsections. Use grammatically normal text instead of separate phrases. Most importantly, define sample plot size and number of repetitions (sample plots) per treatment. Do not use redundant capitalization (composition, total nitrogen, nitrogen). What exactly is meant by "parcel"? Nothing is mentioned on sowing procedures, including methods, doses, timing etc.

In 4.8., indicate how dark septate endophyte structures were distinguished from mycorrhizal structures.

What was source of fungi used in 4.11?

Results

Use past tense exclusively within this part.

Lines 135–138, there is no need to indicate all mycorrhization values from Figure 1 in the text. What is a meaning of brown dashed line in Figure 1, indicated as "%(RLC)", are the data only from experimentally inoculated plants? But what was mycorrhization level in control plants? There is absolutely no reason to claim that "growth trend of the plants was influenced by the development of mycorrhizal colonies inside the root", as growth trends are rather similar for NM vs M plants. From a biological point of view, it is well known that mycorrhizal intensity increases with plant age. Moreover, there is no evidence on any statistically significant differences between the treatments. By the way, is this graph from nutrient-deprived plants?

Lines 162–171, this paragraph (even without any references) is not appropriate for the Results section, it can be incorporated in discussion.

Line 174, it is indicated that plants were grown at optimum fertilizer level ("not limiting for plant growth"), but this is in clear contradiction to the title of the paper stating "in a soil of low fertility". Now I am reading that plants non-inoculated with mycorrhiza received twice as much N (2 Q per ha) as inoculated plants (1 Q per ha), but now I go to 4.2. and see that there are 1.8 Q per ha for NM and 0.9 Q per ha for M, which is not the same. And I still have no idea what the "quintal" is. Does it really mean that mycorrhiza-inoculated vs non-inoculated plants received different doses of fertilizers? Why? This excludes any possibility for comparison between the two on grounds of differences in mycorrhizal treatment. As a result, the part on field results is absolute nonsense in the present form. Because of this, I will not analyze further section 2.3. of the results.

Results described in 2.4. in principle have no direct relation with the first part of the study. Table 1 does not provide any evidence on statistical significances of differences between sites and treatments in respect to fungal composition. Further data are largely qualitative or semi-quantitative, adding little value to the study.

Due to already mentioned problems, there is no need to analyze discussion. However, it seems to be rather theoretical. First I was happy to read at the very beginning of the discussion that N fertilizer for inoculated plants indeed was 50% from that for control plants, so my suspicion that the experiment was wrongly performed was correct, but then I read the next sentence about "This reduction of mycorrhization is in accordance with the concept that the development of mycorrhizal colonization is affected by plant nutritional status" (what reduction? where it was shown?) and I realized that I still did not understand anything and gave up with this review.

Author Response

Reviewer 2

Title

Too long and complicated. It is necessary to concentration on the main aspects of the study, related to mycorrhiza/endophytic and pathogenic fungi interaction. Do not introduce abbreviations. Do not use redundant capitalization ("dark septate").

The authors appreciate the reviewer’s “very positive” attitude toward this manuscript, and we appreciate the comments and questions offered by Reviewer 2.

The title has been modified for clarity and brevity.

Abstract

Not well balanced, difficult to understand nature of performed experiment and the main findings. Do not use unintroduced abbreviations (DSE) and do not introduce abbreviations (NR, GS), as these will not been used further in the Abstract. Do not use redundant capitalization as for enzyme names. Explain what type of experiment was performed (field study, followed by some laboratory tests?), what was modified/manipulated during the experiment. Use past tense exclusively when describing results.

The abstract has been revised to more clearly include the nature of the experiment and main findings. Other suggestions of the reviewer have been included in the revised abstract.

The purpose of the work is stated in the first sentence.

As can be seen from the data included in the abstact, at two  different sites in the field, interference of pathogenic and saprophytic fungi was hindered by mycorrhizal roots that present an impediment to antagonistic fungi and fungal pathogens in the plant’s roots.  The present investigation includes both microbiological observation measurements characterizing various aspects of the metabolic state of the plant, focusing on the efficiency of the nitrogen assimilation pathway. Of key importance, the N assimilation pathway is related to  photosynthesis) and bioenergetic metabolism (glycolysis and respiration), because amino acids are the essential constituents for proteins and enzymes (therefore the overall of metabolic processes).

The authors are not aware of any other direct field studies to determine effects of various fungal species affecting N metabolism and plant growth such as reported in the current investigation. The field was chosen as an experimental environment for plant growth to examine complex biological interactions in a natural biological environment and in which microbiological interference with N metabolism of the plant could be examined. In addition to conducting the experiment in the field, the present, complex investigation includes both microbiological and biochemical studies that were performed in the laboratory.

– line 26–27, is it meant that "nutrient supply from mycorrhizal fungi to plant root tissues resulted in increased plant growth"?

As is well reported in the scientific literature regarding mycorrhizae, the authors consider the mycorrhizal symbiosis an enhancement of the plant roots’ability to capture water and macro and micronutrients of the root system.. In addition, the authors are aware of the possibility that the mycorrhizal fungi may produce metabolic effectors that can affect a gene and proteomic level, exerting influences on the growth and yield of plants..

– line 30, what is meant by "endophytic fungal radicals"?

The wording has been changed to read:

 “radicular endophytic fungi”.

Introduction

Starts rather abruptly, as general part is missing, pointing to particular field (practical or fundamental) to which this study is related. Further, the main logical thread of analysis seems to be missing, as the Introduction consists of separate statements without logical connection between them.

The beginning of the introduction has been rewritten in response to Reviewer 2’s concerns. The interactions among fungal species that were not suggested by the hypotheses stated in the introduction were suggested by the results and are described in the discussion. As the authors have already stated and demonstrated by the present investigation, the mycorrhizaa, at at the field sites studied, seem to limit the colonization of endophytic fungi, including some  of a declared pathogenic nature, and the authors consider this function of mycorrhizae to be a defense aspect for the plant.

Not sure if "cytological interactions" is the best term to describe physiological and biochemical complex of interaction occurring on plant/mycorrhiza interface and affecting plant functions and growth (line 41).

Being biochemists, the investigators of this study determined that “cytological interactions” to be the most appropriate term, because the biochemical interactions affected by the fungi occur in the cytosol of the plant cells.

References are completely missing in the 4th paragraph (lines 61–75). Each fact needs to be supported by a reference. Information included is very general, textbook level text. "Trichoderma" (line 62) is misspelled.

The references have been inserted.

The brief notes on the types of biological interaction between microorganisms are the same as those quoted in the references with the same terms of  Igiehon et al. (2018).

Do not use redundant capitalization (dark septate endophytes) (line 76).

The last paragraph is rather incomprehensible, the authors need to sort out the information and to concentration on the aspects leading to the aim of the study.

The aim needs to be reformulated, as it seems that the real emphasis in this study is on fungal interactions, not functional aspects of nitrogen metabolism, and that these two aspects have been combined "by force" not by functional relatedness. May be the main aim initially was towards physiological aspects of nitrogen assimilation in wheat plants due to mycorrhiza, but fungal interactions just appeared during the study and therefore were included?

We appreciate the reviewer’s insights and suggestions. When starting a research activity and it is discovered that the experimental results can be influenced by the interference of other biological entities, this key factor should not be ignored. Therefore the authors devoted time, resources and energy necessary to explain the phenomenon of nitrogen assimilation, including the interference of several biological entities (i.e. various fungi and the plant).

Materials and methods

In the very beginning, it needs to be clearly stated what kind of field experiments as well as laboratory tests were performed to understand rationale/necessity/complexity of the whole study. At present, it cannot be understood if there were treatments with/without mycorrhizal inoculation vs different fertilizer levels.

No observations were made on the field, as can be seen from the Materials and Methods Results sections. The field was  used as an experimental location chosen for the NM and M wheat cultures, utilizing a precise fertilization plan. The analytical results are from investigations carried out in laoratory, using methods clearly indicated in the Materials and Methods section.

Line 492 and further, quintal is not a scientifically legitimate unit of measurement, use SI system units only. Plant scientific name needs to be indicated. Use "cultivar" (cv) instead of "variety" (var.). Indicate that only part of seed material was inoculated with mycorrhizal fungi.

We agree with the reviewer.

The kilogram (kg) weight as a unit of measurement, has been inserted to replace quintal (Q).

The quintal is a unit of mass and it is equal to 100 kg.

var changed with cv.

Do not use redundant capitalization (total organic carbon, cation exchange capacity).

The redundant capitalization has been eliminated.

Line 505, information is incomplete, CEC cannot be "extracted".

See section 4.2 for methods used.

Line 506, explain what is TEA.

Triethanolamine. Has been inserted.

"4.2. Fertilization plan" is very poorly written. Better provide a scheme in a form of a table how different treatments were performed, including both phenological phase and exact timing of treatment. Do not divide this in subsections. Use grammatically normal text instead of separate phrases. Most importantly, define sample plot size and number of repetitions (sample plots) per treatment. Do not use redundant capitalization (composition, total nitrogen, nitrogen). What exactly is meant by "parcel"? Nothing is mentioned on sowing procedures, including methods, doses, timing etc.

“4.2 Fertilization plan” is a typographic error. It should read, “4.3 Fertilization plan”. It has been revised for clarity.

parcel is the same plot word.

The fertilization plan has been put in Table following the indications of the reviewer.

4.3 Fertilization plan

The cultivable area of each of the two sites was 0.25 ha, divided into two symmetrical parcels, one intended for the non-mycorrhized sowing (NM) parcel, other for mycorrhized sowing (M) parcel.

4.4 Sampling, plant growth and mycorrhizal colonization analyses

Samples were taken 90, (beginning of tillering stage), 110 (beginning of stem-elongation stage), 130, 150, 170 and 190 days after sowing (DAS) following treatment of durum wheat seed with mycorrhizae. At each sampling, ten plants for each parcel of experimental land were collected at 10 randomly distributed points. The ten plants were collected and used for analyses.

In 4.8., indicate how dark septate endophyte structures were distinguished from mycorrhizal structures.

has been clarified and inserted in Section 4.7 and 4.8

What was source of fungi used in 4.11?

The phrase has been rewritten :

Unlike the experiments conducted in growth chambers, in which the development of mycorrhizae, or mycorrhization, occurs in roots growing in relatively sterile soil, the experimental conditions in open field may include the presence and possible conditioning of the endophytic fungi affecting the symbiotic action of mycorrhizae.

The source of fungi used for laboratory analysis was the same as was used in the field. The seeds were inoculated with fungal spores provided by Biotech Spa. The mineral source was a nutritional solution with low concentration of inorganic phosphorus (Pi) both for control and treatment without risking spontaneous mycorrhizations. (seeDi Martino et al., 2018).

Results

Use past tense exclusively within this part.

Lines 135–138, there is no need to indicate all mycorrhization values from Figure 1 in the text. What is a meaning of brown dashed line in Figure 1, indicated as "%(RLC)", are the data only from experimentally inoculated plants? But what was mycorrhization level in control plants? There is absolutely no reason to claim that "growth trend of the plants was influenced by the development of mycorrhizal colonies inside the root", as growth trends are rather similar for NM vs M plants. From a biological point of view, it is well known that mycorrhizal intensity increases with plant age. Moreover, there is no evidence on any statistically significant differences between the treatments. By the way, is this graph from nutrient-deprived plants?

Due to technical problems, the spontaneous mycorrhization curve %RLC of NM plants had disappeared;  we have now reinserted it and the graph is more understandable. The brown dashed line in figure 1 represents the %RCL in M roots. What the authors state is that at 190 DAS the average M shoot weight was 30% greater than NM plants and the grain yield of 25% more than NM plants. Apart from the numbers, these are differences that have an obvious biological significance.

Lines 162–171, this paragraph (even without any references) is not appropriate for the Results section, it can be incorporated in discussion.

It was moved to the Discussion section.

Line 174, it is indicated that plants were grown at optimum fertilizer level ("not limiting for plant growth"), but this is in clear contradiction to the title of the paper stating "in a soil of low fertility". Now I am reading that plants non-inoculated with mycorrhiza received twice as much N (2 Q per ha) as inoculated plants (1 Q per ha), but now I go to 4.2. and see that there are 1.8 Q per ha for NM and 0.9 Q per ha for M, which is not the same. And I still have no idea what the "quintal" is. Does it really mean that mycorrhiza-inoculated vs non-inoculated plants received different doses of fertilizers? Why? This excludes any possibility for comparison between the two on grounds of differences in mycorrhizal treatment. As a result, the part on field results is absolute nonsense in the present form. Because of this, I will not analyze further section 2.3. of the results.

To clarify any misunderstanding, there is a vast literature, to which we have also contributed, which shows that in optimal conditions of fertilization, the process of mycorrhizal colonization is poorly stimulated and the RCL percentages are extremely low. From the Introduction: Suppression of mycorrhizae arises from the high concentration of P and from other  mineral nutrients present in the soil. The suppression of mycorrhizae by high P depends on the overall plant nutrient supply. In addition, a limitation of nitrogen in the soil stimulates root colonization by mycorrhizal fungi [6].

In this way an optimal fertilization in the NM parcel, limits to a minimum the spontaneous colonization of the mycorrhizae, therefore, increasing and not flattening the differences between control and treated plants. This is the exact opposite of what the reviewer understood.

Results described in 2.4. in principle have no direct relation with the first part of the study. Table 1 does not provide any evidence on statistical significances of differences between sites and treatments in respect to fungal composition. Further data are largely qualitative or semi-quantitative, adding little value to the study.

Due to already mentioned problems, there is no need to analyze discussion. However, it seems to be rather theoretical. First I was happy to read at the very beginning of the discussion that N fertilizer for inoculated plants indeed was 50% from that for control plants, so my suspicion that the experiment was wrongly performed was correct, but then I read the next sentence about "This reduction of mycorrhization is in accordance with the concept that the development of mycorrhizal colonization is affected by plant nutritional status" (what reduction? where it was shown?) and I realized that I still did not understand anything and gave up with this review.

As mentioned above, the spontaneous mycorrhization curve of the NM plot has disappeared in Figure 1. Figure 1 currently inserted in the text will make everything more understandable.

Reviewer 3 Report

The manuscript describes the effect of mycorrhization inoculum on wheat growth in the field, including fungal population in the root which is quite novel and interesting.

The authors show a convincing improvement of wheat growth and fitness in mycorrhized condition.

However, the manuscript is a bit difficult to follow, with some parts are way too long and with many repetitions of the same things. This really dilute the interesting message of the paper and need a lot of re-writing.

Also the authors kind of miss the point several times by focusing on the fact that some of their measured proxy are higher or lower in roots than in shoot rather than focusing on the mycorrhization effect. Maybe I did not understand but why (shoot vs root) is that interesting and more importantly how can you explain that?

The analysis of endophytic fungal community is a bit biased as they use PDA culture to reveal the species present in the roots. Hence any species not growing on PDA will be disregarded in this study. A full transcriptomic approach of the fungal community present in the root would be more robust and less biased. But I understand that may represent a financial limit for the team.

It would be nice to note the potential bias in the discussion anyway.

A major issue is that the authors don't measure the mycorrhization rate in the "non-mycorrhized" conditions. Mycorrhizae are present everywhere in the wild and I would be very surprised if the NM conditions would not be a bit mycorrhized.

Please measure this parameter and include it in the manuscript. It will be probably a low mycorrhization rate but it is important to know.

In General, please don't refer to zone 1 and zone 2 in the manuscript. It is very confusing as you switch theses term with M and NM randomly. Only M and NM should be used.

If i understood well zone 1 in NM and zone2 is M? I could only understand that from the M&M.

Please change in the entire manuscript and the figures

Also your data are really convincing and quite clearly presented. You could maybe limit the detailed % highly present in the text. This really adds too much detail not always useful as all the nice info are in your graphs.

L182-184

In leaves, the NRA percentage increase of M plants, compared to NM plants between B1 and B2 and A1 and A2 was about 55% and 40%, respectively.

Simplify like:

In both locations, and in both tissues (leaves and root) we detected a similar NRA increase in M plants compared to NM (fig2a).

Or

In both locations, we detected a similar NRA increase in M plants compared to NM, of 50% in roots and 70% in leaves (fig2a).

Other points:

L19 define DSE

L25 orn ?

L27 guaranteed

L30 radical

M&M:

L487 what varieties ? please put Var. Iride at the very beginning of the paragraph

L516 what pot ? no pot in the study

4.7 and 4.9 are the same no ?

L139 (

2.2 I don't really understand why the authors focus so much on the fact that GSA or NRA is higher or lower in root vs leaves. How is that important? please specify. Maybe i missed a point

What is important is the effect of Mycorrhization on the NRA and GSA.

Moreover, the statistical analysis seems to have been performed on the different datasets (leaves and root) separately. Hence you need to make a global graph showing the statistical significance between leaves and root NRA and GSA, if you want to say it is different in the text.

L163 replace the element by this element

L162-171: not really useful here, should be in intro or removed

L173-174: why difference of fertilizer between the treatments?

Fig2: make simple color or no color, c and d should have same y axis if we want to compare shoot and roots.

L185 already said

What is important is that M plants have higher GSA and NRA in every conditions. The authors should focus on that and simplify this paragraph in like 3 sentences.

Be efficient and straight to the point.

2.4 here the authors mostly talk about differences between sites but not about the potential influence of mycorrhization on fungal population. ex Fusarium avenaceum presence was highly reduced by mycorrhization in both sites. Same for Fusarium equiseti

That is interesting much more than what you find in the different sites. Except if you want to focus on the specie ecological systems in different environment.

Table1 Fig 4: if i understood well A2 and B2 are mycorrhized and A1 B1 are non-mycorrhized. Please names them like that on the figures not 1 or 2. It is quite confusing.

Fig4 is not using the right scaling. We cannot see subtle differences and the missing data are with the same color as the low data (ie Fusarium chlamydosporum is not present if mycorrhized in site A), on the heat map we don't see that.

You should use a gradient of a single color like from light blue to dark blue, and the missing values are in white.

Plus explain more the ◊◊◊◊ * I don't really understand.

3.1 repeat of 2.1, except the very end of it.

Discussion is too long and repeat a lot of the results. I don't think you need a separate section for all the analysis. Just merge to really show your cool results that M improve plant growth.

Your data on A. tellustris are super cool. I hope you will try more experiment on this strain in the field for another publication.

Author Response

First of all, we thank the 3rd reviewer for the scrupulous correction and constructive comments that have improved the paper.

Thanks again

Reviewer 3

The manuscript describes the effect of mycorrhization inoculum on wheat growth in the field , including fungal population in the root which is quite novel and interesting.

The authors show a convincing improvement of wheat growth and fitness in mycorrhized condition.

However, the manuscript is a bit difficult to follow, with some parts are way too long and with many repetitions of the same things. This really dilute the interesting message of the paper and need a lot of re-writing.

Also the authors kind of miss the point several times by focusing on the fact that some of their measured proxy are higher or lower in roots than in shoot rather than focusing on the mycorrhization effect. Maybe I did not understand but why (shoot vs root) is that interesting and more importantly how can you explain that?

The authors agree with the reviewerthat detailed data regarding   amino acids, which are the metabolic fruit of nitrogen assimilation, can be of minimal interest to some readers. However amino acids are key to understanding effects of dark septate fungi and mycorrhizae on nitrogen assimilation and growth of the plant, so we have in included measurements of growth that is affected, in part, by nitrogen metabolism. Amino acids are the key words in molecular language that is established in a symbiotic relationship. Because amino acids are the essential constituents of enzymes, amino acids are therefore, intimately involved inboth  direct and indirect control of metabolic pathways. For this reason, variations of the amino acid patterns provide  indications regarding the metabolism and health  of the plant, as well indicating  possible stress to the plan. The current study includes both the roots and the aerial part of the plant in order to determine metabolic differences between a colonized organ, namely the roots, and the non-mycorrhized leaves. The results we report bring to light significant differences between the reduction of nitrate and the assimilation of ammonium in the two organs of the plant and allow us to reach the conclusions we have stated in the Discussion and Conclusion sections.

The analysis of endophytic fungal community is a bit biased as they use PDA culture to reveal the species present in the roots. Hence any species not growing on PDA will be disregarded in this study. A full transcriptomic approach of the fungal community present in the root would be more robust and less biased. But I understand that may represent a financial limit for the team.

It would be nice to note the potential bias in the discussion anyway.

The authors agree with reviewer 3’s observation. The experiment was designed to  to study the assimilation of nitrogen in mycorrhized plants compared to the unmicorrhized control plants. However,  during the determination of the percentage of mycorrhization under the microscope spontaneous colonization of fungal endophytes like DSE was observed. Not being able to ignore what had been observed, the authors investigated the possible interference between endophytic fungi and the host plant. The exploratory analysis includes what could be considered preliminary fungal screening of the roots, and theauthors understand that a thorough investigation of the entire fungal community in the field soil studied would  require a more detailed, time-consuming and costly in-depth study.

A major issue is that the authors don't measure the mycorrhization rate in the "non-mycorrhized" conditions. Mycorrhizae are present everywhere in the wild and I would be very surprised if the NM conditions would not be a bit mycorrhized.

True, the authors realize that the mycorrhizal curve of the untreated roots, whether it were to show evidence of measurable mycorrhizae or not, is missing.
This issue was included in the discussions, the data was not presented in the graph due to technical inconvenience.

We fixed it. The curve is visible.

In General, please don't refer to zone 1 and zone 2 in the manuscript. It is very confusing as you switch theses term with M and NM randomly. Only M and NM should be used.

The authors understand the difficulty that the use of two sites, namely site A and Site B, may present to some readers; howeverbetween NM and M treatments at each site there are metabolic differences in nitrogen assimilation, regardless of the site of cultivation. The data of Table 1 indicate that between sites A and B the communities of spontaneous endophytes are not identical. The soil microflora in the 2 sites even at short distances differed, and for this aspect of two different sites is presented in connection to the M (2) and NM (1) factors.

If i understood well zone 1 in NM and zone2 is M? I could only understand that from the M&M.

Please change in the entire manuscript and the figures

In Table 1, site and treatment are indicated by a letter and a number. The letters A and B indicate sites A and B, respectively, and the numbers 1 and 2 indicate the control (NM) and the micorrhizal treatment (M), respectively. These designations have been clarified in the text and the figures.

Also your data are really convincing and quite clearly presented. You could maybe limit the detailed % highly present in the text. This really adds too much detail not always useful as all the nice info are in your graphs.

Yes, we will reduce on things that do not give useful information.

L182-184

In leaves, the NRA percentage increase of M plants, compared to NM plants between B1 and B2 and A1 and A2 was about 55% and 40%, respectively.

Simplify like:

In both locations, and in both tissues (leaves and root) we detected a similar NRA increase in M plants compared to NM (fig2a).

To clarify, the text has been modified thusly:

The largest increase in NRA activity of M plants on NM plants was observed in the roots at both field sites with an average of 70% increase (Figure 2 a). A similar pattern of NRA was found in the leaves, but to a lesser extent at both field sites with an average of 45% increase (Figure 2b). It is particularly interesting to note that in NM plants the specific activity of NR was greater in the leaves than in roots. On the contrary, in M plants, the percentage increase in NRA and its absolute values were higher in roots than in leaves.  In the roots, in contrast to NR specific activity, the changes of GS specific activity of M over NM plants in both locations were less than 40% and 35%, respectively. In the leaves, the activity was significantly higher in the M than in the NM plants, where the percentage of activity increase of M over NM plants in both locations between were about 65% . This shows that in M plants, the specific activity of GS was greater in the leaves than in roots both in percentage increase and absolute value.

L19 define DSE

has been done

L25 orn ?

Chemical abbreviation of Ornithine

L27 guaranteed

has been done

L30 radical

has been done

L487 what varieties ? please put Var. Iride at the very beginning of the paragraph

has been done

L516 what pot ? no pot in the study

In the text was written plot no pot which is the equivalent of parcel

However it has been changed to parcel to unify the words of the text.

4.7 and 4.9 are the same no ?

It is the same inserted 2 times

has been deleted

don't really understand why the authors focus so much on the fact that GSA or NRA is higher or lower in root vs leaves. How is that important? please specify. May be i missed a point.

What is important is the effect of Mycorrhization on the NRA and GSA.

NRA and GS are two key and highly regulated enzymes in the process of  nitrogen assimilative, nitrate reduction, and GLU synthesis from which all AA by transamination derive. This process occurs in both the roots and the leaves. In the mycorrhized roots, thanks also to the NRA activity of the fungus, nitrate reduction is elevated with the result that in the mycorrhized roots there is more ammonium than the control roots. In mycorrhized roots  more nitrogen is also in reduced form that is translocated to the leaves where the cytosolic and plastidic GS, with abundance  of carbon skeletons resulting from photosynthesis, has no difficulty in incorporating the reduced N as amino-N in amino acid, glutamine. The result of all this is that the presence of mycorrhizae indirectly increases the amino acid (and therefore protein) pool of the whole plant with other indirect metabolic benefits resulting. Therefore, mycorrhizae tend to polarize the 2 stages of the process, namely the  assimilative reduction of nitrate in the root and the  incorporation of ammonium in amino acids in the leaves. After all, the mycorrhized root, as is  mentioned in the Discussion, by having to host the mycorrhizae, will have to ration the carbon and oxygen reserves for the mitochondria. This is why the synthesis of glutamine is relegated more to the leaves. Moreover, we observed in the root a marked increase of Ala, an amino acid derives from pyruvic acid, the final product of glycolysis.

Moreover, the statistical analysis seems to have been performed on the different datasets (leaves and root) separately. Hence you need to make a global graph showing the statistical significance between leaves and root NRA and GSA, if you want to say it is different in the text.

The authors considered this, as well. The thing is easily solvable and clear graphical reading for the 2 enzymes NRA and GS , becomes more complicated and difficult to interpret for a forest of amino acids. Moreover, the levels of amino acids in plants are the product of enzymatic activities that refer specifically to the NRA and GS.

L163 replace the element by this element

has been done

L162-171: not really useful here, should be in intro or removed

has been done

L173-174: why difference of fertilizer between the treatments?

The different fertilizations aim at testing the hypotheses that:

1) By reducing macronutrients such as phosphorus and nitrogen, a) mycorrhized plants do not experience nutritional deficiency and b) organic mass growth is greater for mycorrhized than than for non-mycorrhized plants.

2) High levels of phosphorus and nitrogen in the soil inhibit the colonization of the fungus. The literature, to which we also contributed, (Di Martino et al 2018) provides strong evidence for this hypothesis.

Fig2: make simple color or no color, c and d should have same y axis if we want to compare shoot and roots.

has been done

L185 already said

What is important is that M plants have higher GSA and NRA in every conditions. The authors should focus on that and simplify this paragraph in like 3 sentences.

Be efficient and straight to the point

The authors  have already rewritten the above hopefully more clearly and synthetically.

2.4 here the authors mostly talk about differences between sites but not about the potential influence of mycorrhization on fungal population. ex Fusarium avenaceum presence was highly reduced by mycorrhization in both sites. Same for Fusarium equiseti

That is interesting much more than what you find in the different sites. Except if you want to focus on the specie ecological systems in different environment.

You are absolutely right! An extra paragraph (3.4) has been inserted in the discussion:

3.4. Fungal screening in roots of wheat plants and possible antagonism  between mycorrhizal fungi  and pathogenic endophytes.

An analysis to explore indigenous endophytic fungi in mycorrhized and non-mycorrhized  durum wheat roots enables one to recognize indicators of healthiness and   pathogenicity and reveals the potential ability of endophytes to colonize roots in competitive relationships with mycorrhizal fungi in soil intended for agricultural crops. At both sites, Fusarium avenaceum and Fusarium equiseti are  the fungi with the highest rate of frequency in the NM roots with average values 85% and 60%, respectively.  They  can be considered to be the most common common species of Fusarium on cereals in temperate climates, and Fusarium fungi are generalist pathogens responsible for diseases in numerous crop species. Fusarium spp. produce mycotoxins which are factors of pathogenicity and virulence in various plant-pathogen interactions. Macrophomine phaseoline was also found in the NM roots of both sites with significant frequencies and average values of approximately 60%. This necrotrophic fungus is an important pathogen of many crops, such as corn, sorghum, potato, and soy.

The fungal screening performed on samples from the 2 sites indicated an interesting result that suggests the potential influence of mycorrhization on the fungal population.  The presence of Fusarium avenaceum, F. equiseti and Macrophomine phaseoline was strongly reduced by mycorrhization at both sites.

Table1 Fig 4: if i understood well A2 and B2 are mycorrhized and A1 B1 are non-mycorrhized. Please names them like that on the figures not 1 or 2. It is quite confusing.

You’re correct. 1 and 2 have been replaced with NM and M throughout the paper.

Fig4 is not using the right scaling. We cannot see subtle differences and the missing data are with the same color as the low data (ie Fusarium chlamydosporum is not present if mycorrhized in site A), on the heat map we don't see that.

You should use a gradient of a single color like from light blue to dark blue, and the missing values are in white

We tried a gradient of a single color, but the effect is worse.

Plus explain more the ◊◊◊◊ * I don't really understand

these indicators identify different strains of the same species

3.1 repeat of 2.1, except the very end of it.

3.1 has been rewritten. 

3.1. Root mycorrhization and plant growth

Durum wheat seeds treated with a mixture of fungal spores Glomus mosseae, G. intraradices and G. coronatum caused the mycorrhization of the roots of durum wheat seedlings in the open field where P availability was limiting and N fertilization was reduced to 50% i.e. 0.9 Q ha-1) compared to traditional treatments. The % RLC began to be significant only after 90 DAS at the beginning of stem elongation. Root samples obtained between 90 and 150 DAS showed hyperbolic growth in the percentage of mycorrhization reaching a plateau to about 50% for both sies. In contrast, the percentage of spontaneous mycorrhization in roots not cholinerated does not exceed 100 DAS 15% both due to the low microbiological load of the natural environment and the robust fertilization to which the non-mycorrhizal plants were treated. This reduction of mycorrhization is in accordance with the concept that the development of mycorrhizal colonization is affected by plant nutritional status and, more exactly, mycorrhization is reduced to in response to greater  availability of macronutrients, in particular phosphorus and nitrogen, for the plant . [34, 35].  The mycorrhizal plants, compared to the non-mycorrhizal plants of the control, showed different growth responses: for both sites the growth of mycorrhizal plants was initially slower, compared to control plants between 90 and about 130 DAS, suggesting part of the energy resources of the plant are destined for the fungal colonization process which slows down its growth. Subsequently, the plants treated with mycorrhizae showed an acceleration in the growth phase up to 150 DAS where the plants were strongly stimulated by the process of nitrogen assimilation,  producing a significant increase in the amino acid pool that enhanced the overall, basic metabolism. These increases in metabolism were indirectly observed in the macroscopic evidence of about a 30% increase of fresh weight. 

2.4 here the authors mostly talk about differences between sites but not about the potential influence of mycorrhization on fungal population. ex Fusarium avenaceum presence was highly reduced by mycorrhization in both sites. Same for Fusarium equiseti

That is interesting much more than what you find in the different sites. Except if you want to focus on the specie ecological systems in different environment.

You are absolutely right! An extra paragraph has been inserted in the discussion:

3.4. Fungal screening in roots of wheat plants and possible antagonism  between mycorrhizal fungi  and pathogenic endophytes.

An analysis to explore indigenous endophytic fungi in mycorrhized and non-mycorrhized  durum wheat roots enables one to recognize indicators of healthiness and   pathogenicity and reveals the potential ability of endophytes to colonize roots in competitive relationships with mycorrhizal fungi in soil intended for agricultural crops. At both sites, Fusarium avenaceum and Fusarium equiseti are  the fungi with the highest rate of frequency in the NM roots with average values 85% and 60%, respectively.  They  can be considered to be the most common common species of Fusarium on cereals in temperate climates, and Fusarium fungi are generalist pathogens responsible for diseases in numerous crop species. Fusarium spp. produce mycotoxins which are factors of pathogenicity and virulence in various plant-pathogen interactions. Macrophomine phaseoline was also found in the NM roots of both sites with significant frequencies and average values of approximately 60%. This necrotrophic fungus is an important pathogen of many crops, such as corn, sorghum, potato, and soy.

The fungal screening performed on samples from the 2 sites indicated an interesting result that suggests the potential influence of mycorrhization on the fungal population.  The presence of Fusarium avenaceum, F. equiseti and Macrophomine phaseoline was strongly reduced by mycorrhization at both sites.

Table1 Fig 4: if i understood well A2 and B2 are mycorrhized and A1 B1 are non-mycorrhized. Please names them like that on the figures not 1 or 2. It is quite confusing.

You’re correct. 1 and 2 have been replaced with NM and M throughout the paper.

3.1 repeat of 2.1, except the very end of it.

3.1 has been rewritten. 

3.1. Root mycorrhization and plant growth

Durum wheat seeds treated with a mixture of fungal spores Glomus mosseae, G. intraradices and G. coronatum caused the mycorrhization of the roots of durum wheat seedlings in the open field where P availability was limiting and N fertilization was reduced to 50% i.e. 0.9 Q ha-1) compared to traditional treatments. The % RLC began to be significant only after 90 DAS at the beginning of stem elongation. Root samples obtained between 90 and 150 DAS showed hyperbolic growth in the percentage of mycorrhization reaching a plateau to about 50% for both sies. In contrast, the percentage of spontaneous mycorrhization in roots not cholinerated does not exceed 100 DAS 15% both due to the low microbiological load of the natural environment and the robust fertilization to which the non-mycorrhizal plants were treated. This reduction of mycorrhization is in accordance with the concept that the development of mycorrhizal colonization is affected by plant nutritional status and, more exactly, mycorrhization is reduced to in response to greater  availability of macronutrients, in particular phosphorus and nitrogen, for the plant . [34, 35].  The mycorrhizal plants, compared to the non-mycorrhizal plants of the control, showed different growth responses: for both sites the growth of mycorrhizal plants was initially slower, compared to control plants between 90 and about 130 DAS, suggesting part of the energy resources of the plant are destined for the fungal colonization process which slows down its growth. Subsequently, the plants treated with mycorrhizae showed an acceleration in the growth phase up to 150 DAS where the plants were strongly stimulated by the process of nitrogen assimilation,  producing a significant increase in the amino acid pool that enhanced the overall, basic metabolism. These increases in metabolism were indirectly observed in the macroscopic evidence of about a 30% increase of fresh weight. 

Discussion is too long and repeat a lot of the results. I don't think you need a separate section for all the analysis. Just merge to really show your cool results that M improve plant growth.

Epeated data was eliminated. The discussion is wide, because we cannot neglect any result. This is why we have left the paragraphs, to give more order to the discussion.

Your data on A. tellustris are super cool. I hope you will try more experiment on this strain in the field for another publication.

Certainly, this work was born from an effective collaboration between the teams of Plant Biochemistry and Plant Pathology, with the aim of continuing the investigations also on the research of bioactive and bactericidal molecules produced by endophytes.

Reviewer 4 Report

Please see the attached file with comments.

Author Response

Reviewer 4

Suggest reworking the title so that it is not two

It has been rewritten. 

DSE spell out

This has been done.

About changes in approximately

This has been changed.

As recommended by the reviewer, the key words in the title have not been listed.

Key words present in the title have been eliminated.

1° sentence in introduction was reformated and changed

The suppression of mycorrhizae by high P depends on the overall plant nutrient supply.

Maybe reword

It has been rewritten.

In addition, a limitation of nitrogen in the soil also stimulates root colonization by mycorrhizal fungi [6]

Results

This section either could be combined with the discussion section or shortened through out.

Bringing the two sections together could increase difficulties in both writing and reading. Also, to submit a result and not comment on it in the discussion section could disappoint the reader that, as we know, is always interested in knowing data that can be useful in his or her research.

Please make the number of replications used for each analysis.

How were these numbers generated? How many replications? Standard error for values in table?

The questions are now answered in the manuscript.

Statistical indications are now inserted in the table.

Other words scattered in the manuscript and reported by the reviewer have been corrected and replaced.

Round 2

Reviewer 2 Report

The irony and arrogant attitude of the authors is completely out of place and even offensive.

The problem of comprehensibility of the study performed and the results obtained still remains.

A large part of specific objections (redundant capitalization only as an example) were simply ignored.

This is a field study, as opposed to a study in controlled conditions, but it is not clearly written anywhere.

The title is still not grammatically correct and conceptual.